# Selective Code Duplication for Soft Error Protection on VLIW Architectures

**Yohan Ko** [1,*], **Soohwan Kim** [2], **Hyunchoong Kim** [2] and **Kyoungwoo Lee** [2]

1 Division of Software, Yonsei University, Wonju 26493, Gangwon-do, Korea
2 Department of Computer Science, Yonsei University, Seoul 03722, Korea; soohwan.kim@yonsei.ac.kr (S.K.); hyunchoong.kim@yonsei.ac.kr (H.K.); kyoungwoo.lee@yonsei.ac.kr (K.L.)
* Correspondence: yohan.ko@yonsei.ac.kr

**Abstract:** Very Long Instruction Word, or VLIW, architectures have received much attention in specific-purpose applications such as scientific computation, digital signal processing, and even safety-critical systems. Several compilation techniques for VLIW architectures have been proposed in order to improve the performance, but there is a lack of research to improve reliability against soft errors. Instruction duplication techniques have been proposed by exploiting unused instruction slots (i.e., NOPs) in VLIW architectures. All the instructions cannot be replicated without additional code lines. Additional code lines are required to increase the number of duplicated instructions in VLIW architectures. Our experimental results show that 52% performance overhead as compared to unprotected source code when we duplicate all the instructions. This considerable performance overhead can be inapplicable for resource-constrained embedded systems so that we can limit the number of additional NOP instructions for selective protection. However, the previous static scheme duplicates instructions just in sequential order. In this work, we propose packing-oriented duplication to maximize the number of duplicated instructions within the same performance overhead bounds. Our packing-oriented approach can duplicate up to 18% more instructions within the same performance overheads compared to the previous static duplication techniques.

**Keywords:** soft error; reliability; VLIW; fault-tolerance

## 1. Introduction

With technological advances, soft errors are becoming increasingly critical concerns in computing systems at the early design phase, especially in modern-embedded systems [1,2] and safety-critical systems such as aerospace computing [3,4]. Soft errors are transient bit flips in semiconductor devices caused by electrical noise, external electronic interferences, cross-talk, alpha particles, neutron, cosmic ray, etc. [5]. The soft error rate is exponentially increasing as the critical charge decreases. Critical charge, which is the minimum charge causing soft errors, and soft error rate have an inversely exponential relationship. Critical charges are decreasing due to a shrinking feature chip size and decreasing supply voltage [6]. Thus, the reliability of embedded systems against soft errors is threatened due to the small form factor intensive low-power computing [7].

Protection techniques for safety-critical embedded systems have been proposed even though soft errors are temporary and non-destructive [8]. Information redundancy techniques, such as error detection and correction codes, can protect memory systems with the simple design complexity based on coding theory [9]. Modular redundancy methodologies also have been proposed for protecting sequential elements and combination logic [10]. Modular redundancy techniques duplicate or triplicate modules, and they can detect or correct soft errors by validating results between the original module result and copied one. However, hardware-based protection techniques incur significant overheads in terms of performance, energy consumption, and hardware area since they require additional hard-

ware components such as coding bits (for information redundancy), redundant modules (for modular redundancy), and validation logic elements (e.g., comparator and voter).

Very Long Instruction Word (VLIW) architectures are parallel processors that maximize instruction-level parallelism [11] to improve performance. Furthermore, VLIW architectures allow system designers to manage multiple parameters such as power and performance since they can provide complete control on instruction scheduling at compile time. VLIW architectures have been exploited in state-of-the-art embedded systems such as the Xtensa Processor and Accelerated Processing Unit (APU) [12–14]. VLIW architectures can be used as co-processors [15] that accelerate kernel parts of a program or customized for IP cores for next-generation mobile devices, home entertainment products, and stream processors [16].

Several instructions can be executed in VLIW processors simultaneously, concurrently, in parallel, while conventional processors can execute instructions in sequence. Thus, compilers for VLIW architectures should prepare a fully scheduled set of instructions, and the VLIW architecture itself does not have to do additional dependency checks between instructions during execution. However, there are many unused instruction slots (No OPerations or NOPs) if there is insufficient instruction-level parallelism in applications. According to our preliminary experimental results, almost half of the instructions are NOPs among several benchmarks in our VLIW architectures.

Several compilation techniques for VLIW architectures have been proposed in order to improve the performance or solve the dependency problems, e.g., reducing the number of NOPs or improving the scheduling algorithm [17]. However, the reliability of VLIW architectures has received much attention since VLIW architectures are commonly employed in "safety-critical" embedded systems such as aerospace, automotive, and rail transport applications [18]. In order to improve the reliability of VLIW architectures, several compiler-assisted instruction duplication [19,20] have been proposed by using NOPs and additional code lines. However, all the instructions in applications cannot be duplicated without performance overhead even though the complication techniques use the existing NOPs.

In order to enlarge the number of duplicated instructions, previous complication schemes add NOP instructions intentionally. Assume that all the instructions need to be duplicated for complete protection. Our preliminary experimental results show that the protection can incur up to 67% performance overhead as compared to unprotected cases when we duplicate all the instructions. Therefore, it is inappropriate for resource-constrained embedded systems due to the severe performance overhead. In order to mitigate the performance overhead, original instructions in VLIW architectures can be duplicated under the architect-defined performance boundary, e.g., the number of instructions. The result of the original instruction and duplicated instruction are compared to detect bit flips caused due to soft errors. The compiler-assisted algorithm can protect VLIW architectures with minimal hardware modification to store original and duplicated instructions and validate results between them.

However, previous solutions cannot provide efficient reliability against soft errors since they have no priority-based code duplication algorithms within a given performance bound. They duplicate instructions by the sequential order of applications. In this paper, we propose priority-based instruction duplication techniques for VLIW datapaths. We define the priority of instructions to duplicate more instructions within the architect-defined code line overheads. We suggest a packing-oriented duplication to maximize the number of duplicated instructions compared to previous static schemes. In the packing-oriented duplication scheme, unduplicated instructions in the same cycle are defined as a single pack, and our approach selects packs with more instructions. Our experimental results show that our packing-oriented method can duplicate up to 18% more instructions than the existing static duplication under the same code line overheads without any additional hardware overheads.

The rest of this paper is structured as follows. In Section 2, we have introduced related research regarding soft error protection techniques for embedded systems and VLIW architectures. Our approach and algorithm written by pseudo code are described in Section 3, and experimental setup and observations are summarized in Section 4. Finally, Section 5 concludes this paper and guides the direction of future research.

## 2. Related Works

The reliability against soft errors in embedded processors is becoming a paramount design concern, especially in embedded systems with the development of the technology [1]. Soft errors are temporary bit flips in the semiconductor device by external radiation, alpha particles, neutrons, and even cosmic ray [21]. The soft error rate is continuously increasing even with FinFET technology due to the reduction in the critical charge [22,23], and threats of these rising soft errors cannot be ignored anymore. Moreover, the reliability of embedded systems is becoming more important [1] since embedded systems can be exploited in crucial applications such as fiscal applications, mobile healthcare systems, IoT devices, and even automotive systems in the near future.

Logic elements, such as sequential elements and combinational logic, are one of the most sensitive parts to soft errors [24]. Spatial redundancy-based techniques, such as DMR (Dual Modular Redundancy) and Triple Modular Redundancy (TMR), can improve the reliability against soft errors. DMR exploits one additional module to replicate and detect soft errors by comparison between outputs of the original module and duplicated module. DMR can detect soft errors with relatively less overhead, but it cannot correct soft errors without additional mechanisms, e.g., checkpoint and rollback [25]. TMR has been proposed to correct soft errors by using additional two modules and a majority voting mechanism. In TMR systems, original modules are triplicated into three modules (original and additional two modules) running in parallel, and their results are corrected by majority voting. In order to minimize area overheads from TMR, optimized TMR has been proposed for VLIW architectures [26]. However, modular redundancy schemes incur considerable overheads in terms of hardware area, power, and performance due to the module duplication/triplication and expensive comparison/voting mechanism.

Temporal redundancy-based techniques have been proposed to protect logic circuits. The same circuit operates the same data multiple times. Original results are compared with subsequent results to detect soft errors. In order to correct soft errors, operations should be executed three times, and a two-out-of-three voting mechanism selects correct results. Hyman et al. [27] proposed temporal redundancy by using the Latency Slack Cycle (LSC) of instructions for multi-core processors. LSC is the number of cycles before the computed result from the instruction becomes the source operand of a subsequent instruction. Protection techniques based on temporal redundancy can improve reliability with less hardware area overhead compared with modular redundancy. However, it still incurs severe overhead in terms of performance and hardware area.

For these reasons, hardware-based protection techniques are not appropriate for resource-constrained embedded systems due to the additional hardware modification. In order to minimize hardware modification, various software-based protection techniques have been proposed. Oh et al. [28] proposed EDDI (Error Detection by Duplicated Instructions), which duplicates instructions at different registers or variables and inserts a comparison and validation at store or branch operation during the compile stage. EDDI can detect most errors in common except control flow check without any hardware modification or additional area overhead. However, EDDI induces performance overhead since it duplicates and compares all the instructions without the priority of instructions.

Compiler-assisted instruction duplication techniques are appropriate for VLIW architectures since all the instructions in VLIW architectures can be scheduled at the compile stage. Holm et al. [29] proposed a software-only fault-tolerant technique for VLIW architectures by inserting redundant instructions into functional units. Bolchini et al. [30] presented additional instructions to detect hard and soft errors for a VLIW datapath. How-

ever, complete duplication of all instructions can incur severe overhead in terms of energy and performance due to a large number of instructions. In order to reduce the overhead from software-only protections, Hu et al. [19], Lee et al. [20], and Sartor et al. [31] proposed a VLIW scheduling algorithm by exploiting unused instruction slots with minimal hardware modules such as register queues and comparator. Their approaches can schedule the original instructions and its duplicated instructions with considering performance or energy boundary.

However, existing approaches did not define nor consider the priority of instructions, so they duplicate instructions by the sequential order of instructions. In this paper, we define the importance of instructions to duplicate instructions effectively within the same overhead as compared to previous static schemes.

### 3. Our Approach

#### 3.1. Enhanced VLIW Architecture for Error Protection

In this paper, we assume that our VLIW datapath is composed of two integer ALUs, one multiplier, and one load/store unit, and one branch unit as illustrated in Figure 1. Note that we only use integer operation as the target code of VLIW architecture. Since our VLIW architecture has no branch predictor, branch instructions in our architecture work the same way as compared to other integer operations. In order to detect soft errors, comparator (cmp) and register queues (IRVQ—Integer Register Value Queue and LSAQ—Load/Store Address Queue) are needed. After the original and its duplicated instructions are processed, computation results are stored in corresponding register queues. If results from both original and duplicated instructions are stored in register queues, result values from register queues are compared by comparators as shown in Figure 1. Soft errors can be detected if the results of original and duplicated instructions are different. note that our VLIW architecture consists of the comparator and functional units. If the functional unit executes the redundant instructions, we compare the results from the original and duplicated instructions. If they are not identical, it is defined as soft errors on the datapath. Note that we have modified ISA [20] to recognize whether an instruction is or not. If an instruction is not duplicated, our architecture skips the comparison process.

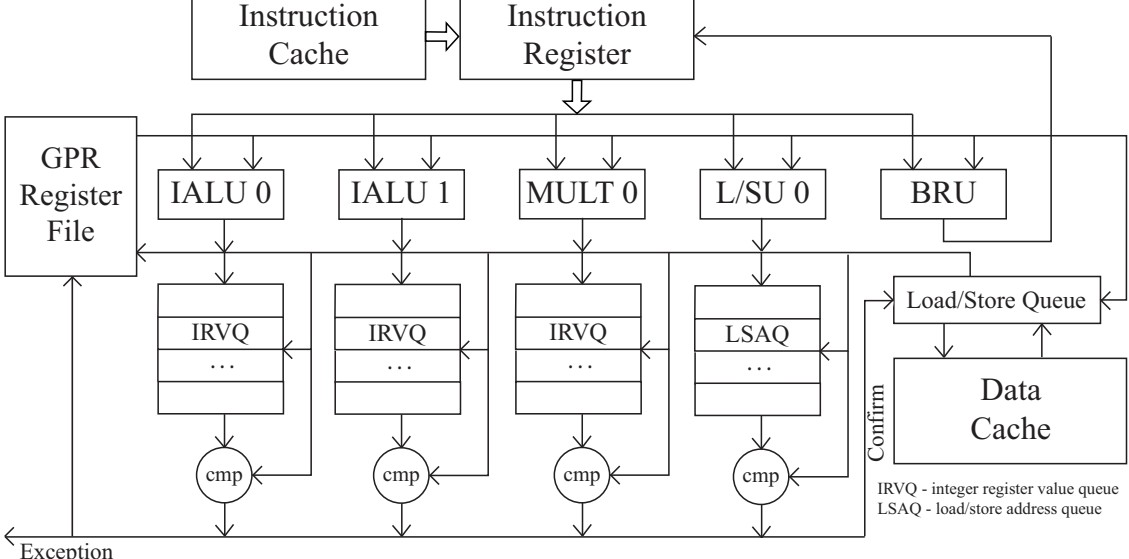

**Figure 1.** Our VLIW architecture needs additional queues to store temporal results from original and redundant instructions (e.g., IRVQ and LSAQ), and it also needs comparator (cmp) to detect soft errors.

*3.2. Selective Code Duplication with Comparable Fault Coverage*

The previous static duplication decides the duplicated instruction according to the sequential order of scheduled instructions. In order to increase the number of duplicated instructions, we propose packing-oriented duplication. The more number of instructions are duplicated, the better we can achieve reliability against soft errors. For instance, the static duplication can duplicate just 39% out of the total instructions without additional code lines based on our preliminary experimental results for a benchmark convolution. Thus, additional cycles are required to duplicate the remaining 61% instructions. In this case, the static duplication adds additional cycles in order of scheduled instructions without considering the number of duplicated instructions.

```
A: MUL r2, r7, r8
B: SUB r1, r7, r8
C: LOAD r8, r2
D: MUL r7, r1, r3
E: ADD r4, r2, r3
F: ADD r5, r2, r6
```

In order to describe the processes for instruction duplication, we have chosen a set of code segments including integer ALU operation (ADD and SUB), multiplication (MUL), and memory (LOAD) operations with considering register dependencies among operations as shown in Figure 2. In Figure 2, $rN$ indicates an N numbered register in register files. The first operand is the destination, and the second and third operands are the sources. For example, operation A: MUL $r2$, $r7$, $r8$ executes a multiplication of contents in $r7$ and $r8$ and stores the result in $r2$. In the case of memory instructions, they use the first and second operands as the register index and memory address, respectively. For instance, operation C: LOAD $r8$, $r2$ accesses the memory address using $r2$, and the value in the memory address writes the value of $r8$.

Our original compiler schedules the code segment and generates the scheduled instruction sequences for VLIW architecture as shown in Figure 2a. The scheduled instructions in Figure 2a present two empty slots in our 4-way VLIW datapath that are unused each cycle, i.e., NOPs. Thus, the compiler can duplicate only one instruction (instruction B) out of six instructions without the additional code lines as shown in Figure 2b. If the additional one code line is permitted by system engineers, just one more instruction (instruction A) can be duplicated at cycle 1 as depicted in Figure 2c. It is because that the static duplication adds the cycle to duplicate as following the order of instructions. As a result, only one instruction of the remaining five instructions (20%) is duplicated with one code line overhead in this scenario.

In our packing-based duplication, the compiler adds the code line to duplicate more instructions than the previous static duplication. The unduplicated instructions at the same cycle are packed and considered as a unit. In Figure 2a, only instruction A is in a pack at cycle 0 since instruction B can be duplicated at the same cycle. In cycle 1, instruction C, D, E, and F are considered a single pack. In other words, the pack size of cycle 0 is only 1, while that of cycle 1 is 4. Thus, cycle 1 is selected to be duplicated by our packing-based duplication, and four instructions out of the remaining five instructions (80%) can be duplicated within one code line overhead as shown in Figure 2d. In this simple scenario, our packing-based duplication increases the number of duplicated instructions by 60% as compared to the existing static duplication. Note that the original scheme duplicates just 20% of the remaining instructions, while our packing-based duplication duplicates 80% of them.

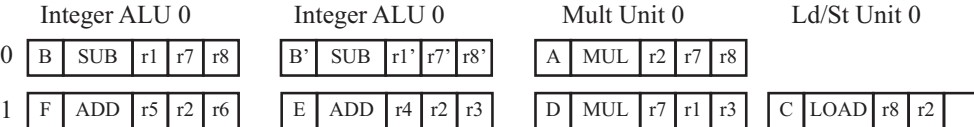

(**a**) Original instructions scheduled on our VLIW architecture.

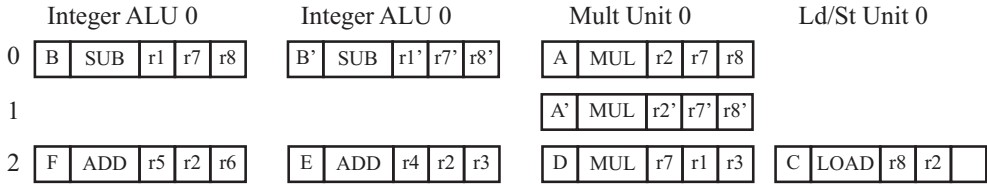

(**b**) Instruction duplication without performance overhead. Only 1 instruction can be duplicated.

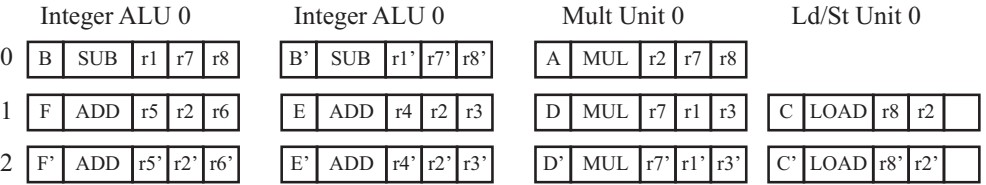

(**c**) Previous static instruction duplication with 1 line overhead. Only 1 instruction out of the remaining 5 (20%) can be duplicated.

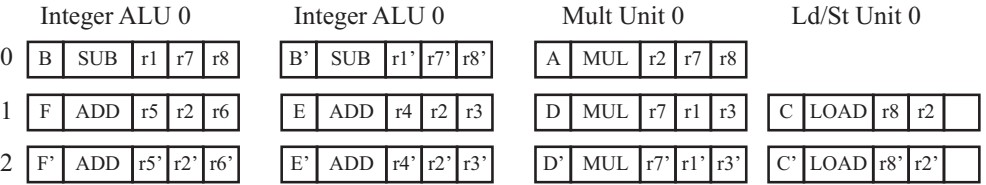

(**d**) Our packing-based instruction duplication with 1 line overhead. Out of remaining 5 instructions, 4 instructions (80%) can be duplicated.

**Figure 2.** Simple scenario to show the efficacy of our packing-based duplication scheme. Within the same code line overhead (1 line), our packing-oriented scheme can duplicate 60% more instructions.

### 3.3. Profiling-Based Algorithm for Packing-Oriented Duplication

In order to predict the number of duplicable instructions, we have implemented the profiling-based algorithm as described in Algorithm 1. Assume that the original assembly code consists of 9 instructions such as A1, B1, A2, B2, D2, A3, D3, A4, B4, C5, and D5, and its runtime is 5 cycles as shown in Figure 3. In Figure 3a, prefix A and B represent multiply and subtraction, respectively. The prefix C and D are memory and multiply instructions, respectively. The number in the postfix is the cycle, so A1 represents ALU instruction which runs on the first cycle.

First off, our algorithm duplicates instructions that do not require additional instructions. Line 7 and 8 copies ALU instructions within the same cycle as described in Algorithm 1, and we also copy instructions in the different cycle using NOP instruction. Our algorithm counts the number of unduplicated instructions (UnDup) as shown in Line 10, 11, and 14, and we also count the number of duplicated instructions (DupOther) in other cycles in Line 14. For instance, there are no unduplicated instructions (UnDup: 0) in the first cycle since A1 and B1 are duplicated in the fifth cycle (DupOther: 2). In the case of the third cycle, UnDup is 1 (D3) instead of 2 since the original instruction (A3) is duplicated at the same cycle. Without performance overheads, 7 shaded instructions cannot be duplicated as shown in Figure 3b.

---

**Algorithm 1** Profiling-based algorithm to predict the number of duplicable instructions

---

1: **procedure** PROFILING-BASED ALGORITHM($n, c$)
   ▷ Parmeters n: number of total instructions, c: number of additional instructions
2:     UnDup[n] = {0}
3:     DupOther[n] = {0}
4:     Sum[n] = {0}
5:     **for** i = 1 to c **do**
6:        **for** j = 1 to n **do**
7:           **if** (unit[0] != NOP) OR ((unit[1] == NOP) **then**
      ▷ Both unit[0] and unit[1] execute ALU instrutions
8:             duplicate unit[0] to unit[1]
9:           **end if**
10:           UnDup[j] = (number of instructions)
11:           UnDup[j] -= (duplicated instructions at the same cycle)
12:           **for** k = i to n **do**
13:             UnDup[j] -= (duplicated instructions at the $k$th cycle)
14:             DupOther[j] += (duplicated instructions at the $k$th cycle)
15:           **end for**
16:           Sum[j] = UnDup[j] + DupOther[j]
17:        **end for**
18:        Target = Select the index which has the largest number from *Sum* array
19:        Add NOP cycle after line *target*
20:        Copy instructions by using additional NOP cycle
21:     **end for**
22: **end procedure**

---

(**a**) Original instructions without duplication.

(**b**) Instruction duplication without performance overheads.

(**c**) Instruction duplication based on the sequential order of instructions.

(**d**) Instruction duplication based on our profiling-based algorithm.

**Figure 3.** Our profiling-based algorithm can predict the number of duplicable instructions based on the number of unduplicated instructions and duplicated instructions at different cycles.

In order to maximize the number of duplicated instructions with the performance overhead, we choose the instruction which has the largest sum of `UnDup` and `DupOther` instead of the sequential order of instructions. Figure 3c shows the static scheme which duplicates instructions by the sequential order of instructions. Since the NOP cycle is added after the first cycle, we mark the additional cycle as 1' as depicted in Figure 3c. And, it duplicates 2 more instructions, which is the sum of `UnDup` and `DupOther` at the first cycle, as compared to Figure 3b. Based on the profiling-based approach, we select the second cycle as the duplication target since it has the largest sum of `UnDup` and `DupOther` as shown

in Figure 3d. In this case, it duplicates 3 more instructions as compared to the base case (Figure 3b) since the sum of `UnDup` and `DupOther` at the second cycle is 3. Assume that we add 1 more instruction based on Figure 3d. We select the first cycle as the duplication target since it has the largest sum among the remaining cycles.

Without a profiling-based algorithm to predict the number of duplicable instructions, we have to analyze all the possible cases. If we add 1 cycle based on Figure 3b, we need to analyze 5 cases. This is because that we can add an instruction through the first cycle to the fifth cycle. Assume that we add $n$ cycles of NOP cycles, and its runtime is $c$ cycles. In this case, $_cC_n$ cases are possible since we can select $c$ instructions as the copy target out of total $n$ instructions. For instance, the benchmark convolution consisted of 46 instructions. If we add 10 cycles into this benchmark, we need to analyze $_{46}C_{10}$ (=4,076,350,421) cases. On the other hand, our profiling-based algorithm needs to check just 10 cases since it just counts the number of duplicated instructions as described in Algorithm 1.

## 4. Experiments

### 4.1. Experimental Setup

Figure 4 shows our experimental framework in order to present the efficacy of our packing-based duplication. All the benchmarks in this paper are code segments from DSPstone [32], digital signal processing-oriented applications. Note that code segments are selected in the loop since VLIW architecture is used to accelerate the loop iteration. This is because compilation techniques suppose 10 times of execution for instructions in the loop [33]. Thus, it can improve the overall performance when we accelerate instructions in a loop rather than other instructions since the loop takes up the majority of execution time. The high-level code from these benchmarks is translated to common low-level language via GCC (Version 4.1) at the front-end phase.

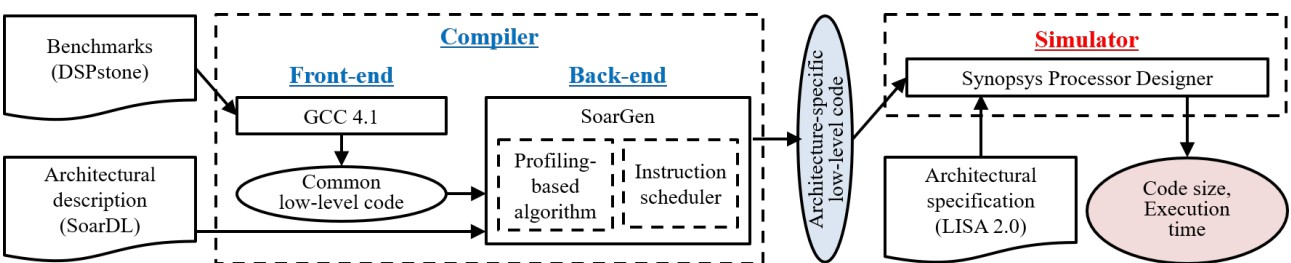

**Figure 4.** Our compiler-simulator framework to present the efficacy of our approach.

At the back-end phase of the complication, the standard low-level code is translated to architecture-specific low-level code using the architecture description language, SoarDL [34]. During the back-end phase, we have modified a retargetable compiler platform, SoarGen [34], to implement our packing-based duplication as described in Algorithm 1. After two-level compilation, the high-level source code can be executed by our VLIW architecture. In order to show the efficacy of our packing-based duplication compared to the static duplication, we have also implemented the instruction duplication in sequential order.

The low-level code is executed on the simulator to estimate the code size and execution time. We have used the Synopsys Processor Designer [35] to generate a software-level simulator to imitate the proposed VLIW architecture. Note that we use 90 nm technology to simulate our VLIW architecture. We selected 4-way VLIW architecture which is described by hardware description language LISA 2.0 [35]. The baseline architecture behind VLIW architecture exploits a standard RISC-style ISA such as MIPS [36]. Our 4-way VLIW architecture is composed of 4-issue slots such as two Integer ALUs (IALU), one Multiplier (MULT), one Load/Store Unit (L/SU), and one Branch Unit (BRU). We estimated the cell area of our VLIW architecture by using the Synopsys Processor Designer [35] to detect soft errors as shown in Table 1. Even though our VLIW architecture needs the comparator to detect erroneous data, it incurs just 3.2% overhead in terms of hardware overhead.

**Table 1.** Cell area of our VLIW architectures which is modified to detect soft errors.

| Type | Original | Packing-Oriented Duplication |
|---|---|---|
| Combinational area | 426,636 μm$^2$ | 437,481 μm$^2$ |
| Non-combinational area | 136,998 μm$^2$ | 144,327 μm$^2$ |
| Total | 563,633 μm$^2$ | 581,808 μm$^2$ |

*4.2. Experimental Results*

Figure 5 shows the efficacy of our packing-oriented protection schemes compared to the static scheme for a benchmark convolution. The vertical axis represents the ratio of duplicated instructions among the entire instructions. The horizontal axis represents the allowed performance overheads in terms of execution time. If the number of additional code lines is zero, we do not allow the performance overhead for duplication. Without adding code lines, about 39% of the entire instructions can be duplicated by exploiting unused instruction slots for a benchmark convolution.

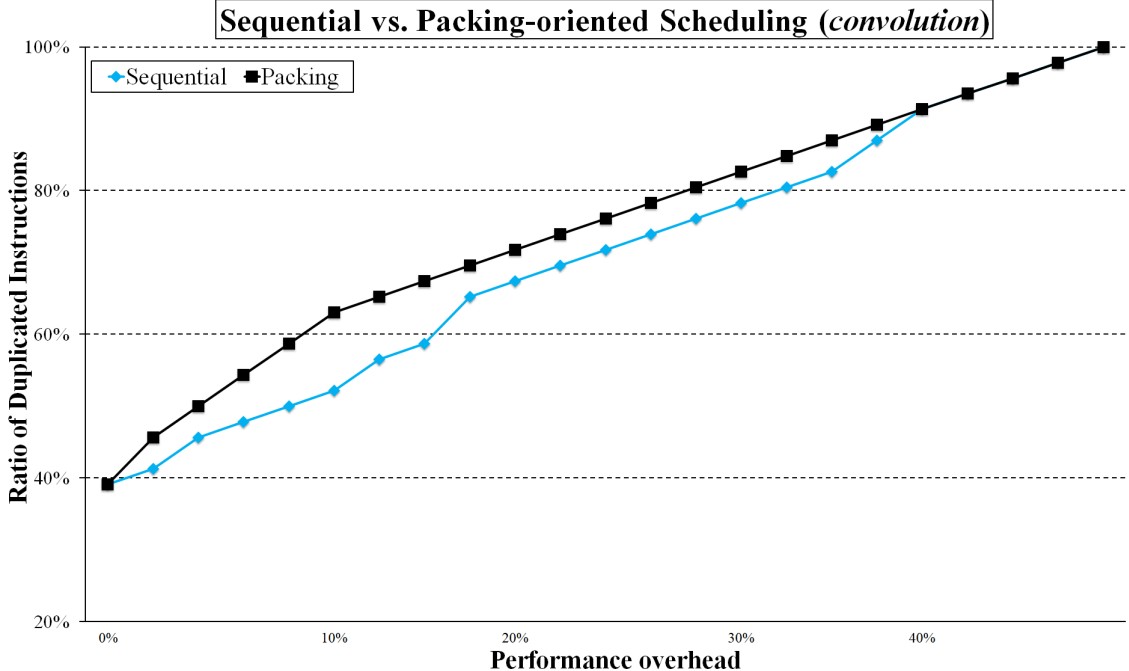

**Figure 5.** Our packing-oriented duplication can duplicate 18% more instructions as compared to previous sequential duplication.

We need to allow more code lines to duplicate the remaining 61% of the entire instructions. In order to duplicate all the instructions for the benchmark convolution, almost half of the execution time is required as shown in Figure 5. If we allow 2% performance overhead for the benchmark convolution, the previous sequential scheme can duplicate only 41% of the remaining instructions. On the other hand, our packing-oriented scheme can duplicate 52% among entire instructions. If 11% of performance overhead is allowed by system engineers, only 64% of the remaining instructions can be duplicated by the sequential scheme, while our packing-oriented scheme can duplicate 39%.

Table 2 shows the maximum difference of duplicated instructions between the sequential scheme and packing-oriented one within the same performance overhead in terms of execution time. For example, a packing-oriented scheme can duplicate 4% more instructions than the previous static approach for a benchmark complex_multiply. Our packing-oriented scheme can duplicate an additional 9% of remaining instructions as compared to the static sequential approach on average.

**Table 2.** The maximum efficiency of our packing-oriented approach as compared to the sequential scheme.

| Benchmarks | Maximum Difference |
|:---:|:---:|
| complex_multiply | 4% |
| complex_update | 12% |
| convolution | 18% |
| dot_product | 14% |
| fir | 6% |
| fir2dim | 3% |
| iir_biquad_one_section | 13% |
| lms | 8% |
| matrix | 3% |
| Average | 9% |

This maximum difference between the static sequential approach and the packing-based one can vary 3% to 18% depending on the characteristics of benchmarks. For example, only 3% of remaining instructions can be copied by our packing-oriented scheme as compared to the static scheme for the benchmark matrix. This is because the beginning sections of this application have a larger number of packed instructions rather than middle or end sections in the case of the benchmark matrix. On the other hand, in the case of the benchmark convolution, the middle of the application has a much larger number of packed instructions than the beginning sections.

## 5. Conclusions

VLIW architectures are deployed in the many modern-embedded systems in order to improve performance, and they are also exploited for safety-critical applications such as healthcare and automotive systems. As the reliability of embedded systems against soft errors is becoming more critical with technology scaling, several techniques have been proposed to protect VLIW architectures. Hardware-based techniques such as modular redundancy incur overheads in terms of area and power consumption because of additional hardware modification, and software-based methods, such as instruction duplication, have a huge performance overhead due to expensive validation. In order to bridge the gap between them, compiler-assisted instruction duplication techniques have been proposed for VLIW architecture. However, the previous static approaches duplicate instructions according to the sequential order of instructions, so they cannot effectively exploit the instruction-level parallelism in VLIW architectures. In this paper, we present the packing-oriented duplication in order to maximize the number of duplicated instructions within the same performance overheads. We define the pack as the number of duplicable instructions in the same code line, and we select larger packs for instruction duplication. Our experimental results reveal that the packing-oriented approach can duplicate 18% more instructions as compared to the static one within the same code line overheads.

**Author Contributions:** Data curation, S.K.; Software, Y.K. and H.K.; Supervision, K.L.; Writing—original draft, Y.K.; Writing—review and editing, Y.K. All authors have read and agreed to the published version of the manuscript.

**Funding:** This work is supported by IITP grant funded by the Korean government Ministry of Science and ICT (MSIT) under the project titled Research on High Performance and Scalable Many-core Operating System (Grant No: 2014-3-00035).

**Conflicts of Interest:** The authors declare no conflict of interest.

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
