# Peer review of "Selective Code Duplication for Soft Error Protection on VLIW Architectures"

_electronics, doi:10.3390/electronics10151835_

Round 1
Reviewer 1 Report
- An unfinished sentence "Further, soft" appeared at the end of the first paragraph in the Introduction section, possibly by editorial error.
- The authors of the revised version of the publication removed many of the mechanisms described in the previous version, which - as indicated in the first review - were not well explained. The authors justified it as follows: "We have updated our paper to eliminate the idea of error correction for better readability since it is not explained in a detailed way. In this paper, our VLIW architecture just detects soft errors ... ”. Thus, in my opinion, the publication presented for review is not valuable enough.
- There are still some inconsistencies in the markings or simply errors at work. For example, the operation "MUL r7, r1, r3" contained in listing on line 167 is described in the text "floating point multiplication", while the rest of the instructions in this listing (i.e., ADD and SUB) perform integer operations on the same registers. eg. on register "r3". This seems to be incorrect. The methods of binary encoding of integer and floating-point numbers are significantly different and it is not possible - without conversion - to mix the two data types. I believe that this part of the work has not been done well enough as well.
Author Response
- An unfinished sentence "Further, soft" appeared at the end of the first paragraph in the Introduction section, possibly by editorial error.
→ I have eliminated the meaningless sentence from the paper. Therefore, this sentence is excluded from the paper by the editorial mistake.
- The authors of the revised version of the publication removed many of the mechanisms described in the previous version, which - as indicated in the first review - were not well explained. The authors justified it as follows: "We have updated our paper to eliminate the idea of error correction for better readability since it is not explained in a detailed way. In this paper, our VLIW architecture just detects soft errors ... ”. Thus, in my opinion, the publication presented for review is not valuable enough.
→ In the previous version of the paper, our paper consists of two main contributions such as error correction and packing-oriented algorithm. However, there is a lack of experimental results for error correction schemes and a detailed explanation of packing-oriented algorithms. The current version of the paper focuses on the packing-oriented algorithm as the main contribution. Thus, the current paper includes the detailed algorithm written in pseudo-code, runtime experimental results, and hardware analysis for implementing additional hardware modules.
- There are still some inconsistencies in the markings or simply errors at work. For example, the operation "MUL r7, r1, r3" contained in listing on line 167 is described in the text "floating point multiplication", while the rest of the instructions in this listing (i.e., ADD and SUB) perform integer operations on the same registers. eg. on register "r3". This seems to be incorrect. The methods of binary encoding of integer and floating-point numbers are significantly different and it is not possible - without conversion - to mix the two data types. I believe that this part of the work has not been done well enough as well.
→ The point from the reviewer is valid. In this paper, we only use the integer operation as the target of acceleration. We have added the footnote in the paper as follows.
“Note that we only use integer operation as the target code of VLIW architecture for simplicity.”

Reviewer 2 Report
I appreciate that the authors have evaluated the hardware cost in Table 1. In addition, I would like to ask two questions as follows:(1) What is the process technology (e.g., TSMC 45nm) used when synthesizing the processors?
(2) How long is the critical path delay (or clock constraint) of the synthesized processors?
Author Response
- What is the process technology (e.g., TSMC 45nm) used when synthesizing the processors?
→ In this work, we use 90 nm when we synthesize the processor. We have added the following footnote in the manuscript.
“Note that we use 90 nm technology to simulate our VLIW architecture.”
- How long is the critical path delay (or clock constraint) of the synthesized processors?
→ In this work, we use 22 MHz as the clock cycle. However, we do not add this setup in this paper since it is beyond our scope.

Reviewer 3 Report
The manuscript is generally well written and organized. Minor comments are listed in the following
In the introduction line 26, there is a broken sentence.
Which is the rationale behind the setup choice in section 3.1? The hardware cost for the comparator and the register queues depends on that choice .
The authors claim that their approach can duplicate 60% more instructions can they show the formula to determine this number (that, the reviewer imagines, depends on the setup)
The profiling-based algorithm presents some possible flaws:
- There is an error at line 7 of the Algorithm (an extra parenthesis) and also the condition should be fixed: (unit[0] != NOP) && (unit[0] == NOP) has no sense.
- UnDup[j] is assigned at both line 10 and at line 11 of the Algorithm. I.e., the value determined at line 10 is not used.
Please check the whole procedure presented in Algorithm1
At line 225 there is a possible typo in c Cn (the first c is superscript)
Author Response
- In the introduction line 26, there is a broken sentence.
→ I have eliminated the meaningless sentence from the paper. Therefore, this sentence is excluded from the paper by the editorial mistake.
- Which is the rationale behind the setup choice in section 3.1? The hardware cost for the comparator and the register queues depends on that choice.
→ In this paper, we only use the integer operation as the target of acceleration. We have added the footnote in the paper as follows.
“Note that we only use integer operation as the target code of VLIW architecture for simplicity.”
Table 1 shows the area overhead caused by the compartor and queues, and it is less 4% as compared to the original architecture, which does not duplicate instructions.
- The authors claim that their approach can duplicate 60% more instructions can they show the formula to determine this number (that, the reviewer imagines, depends on the setup)
→ I have added a detailed explanation as to the footnote into the current version of the paper.
"Note that the original scheme duplicates just 20\% of the remaining instructions, while our packing-based duplication duplicates 80\% of them."
- The profiling-based algorithm presents some possible flaws:
- There is an error at line 7 of the Algorithm (an extra parenthesis) and also the condition should be fixed: (unit[0] != NOP) && (unit[0] == NOP) has no sense.
- UnDup[j] is assigned at both line 10 and at line 11 of the Algorithm. I.e., the value determined at line 10 is not used.
- Please check the whole procedure presented in Algorithm1
→ The condition is fixed as follows. This condition shows the case that the first instruction is not NOP instructions, and the second instruction is NOP.
“(unit[0] != NOP) OR ((unit[1] == NOP)”
In Line 11, we use the minus to get the number of undplicated instructions.
- At line 225 there is a possible typo in c Cn (the first c is superscript)
→ The current version of the paper fixes the typo above.

Reviewer 4 Report
Authors are proposing a compile time mechanism to improve the instruction duplication efficacy in VLIW architectures. Instruction duplication is used to catch soft errors in VLIW architectures where the chance of erroneous data during computation or load store is enhanced because of the decreased critical charge in chips. Their mechanism improves the instruction duplication efficiency of VLIW computers by identifying the cycles in instructions which can yield maximum duplication. They use profiling to identify a set of instruction lines which can yield maximum duplication. Authors have focused on a very specific subsection for optimization.
The paper is well written except for some typos. Authors have clearly explained their algorithm using specific instruction set combinations. They have used a simulator to demonstrate the effectiveness of their mechanism.
comments:
- line 26: seems like an incomplete sentence.
- line 64: 'protection incurs up to 67% of performance overhead'. Can you elaborate why the overhead is 67%?
- Section 3.1. Explain how you skip the comparison when there is no redundancy in instruction executions.
- Line 160 to 162: can you explain how you got the numbers 42% and 58%?
- Line 175: explain how LOAD instruction works.
- Line 200 typo: multiply and memory instructions
- Figure 3. You should include a scenario which contains branch instructions. In general, your analysis doesn’t include any cases of branch instructions.
- Figure 4 label is incorrect.
- Can you elaborate more on line 235-236 as to how the VLIW architecture is used to accelerate loop iteration?
- Section 4.2: you have reversed the H and V axis in description.
- Line 271. 39% is a typo.
- Line 294 - 295. ‘You claim to propose the enhanced VLIW architecture that executes a third instruction’ but your paper only covers the pack based duplication.
Round 2
Reviewer 1 Report
The authors referred to the suggestions expressed in previous reviews and improved their publication. In my opinion, the work is still not very valuable. However, it can be accepted for publication in the journal.
This manuscript is a resubmission of an earlier submission. The following is a list of the peer review reports and author responses from that submission.
Round 1
Reviewer 1 Report
The paper presents a technique for soft error protection targeting VLIW architectures.
The approach is very simple and it is a very limited step forward (if any) in the field. The claimed advantage is mainly in terms of packing-based duplication, which is based on well-known techniques in VLIW that try either to use empty slots to hint at future behaviors or to add instructions.
The description of the proposed approach is not detailed making the story more difficult to capture for the reader.
The authors link their approach with a particular type (in terms of datapath, i.e. execution-L/S units) of VLIW architecture. However, the proposed technique is independent of it, since it uses empty slots in the long instruction.
The description related to Figure 1 does not clarify how the redundant instructions are managed.
Figure 2 can be omitted. Moreover, it shows a classical RISC pipeline, and it is not clear how it is related to the VLIW architecture of figure 1.
Section 3.2. The authors claim some percentage of total instruction that can be duplicated without additional code lines (i.e. 42%) without justifying the origin of the number. Is it an average value on what?
The experimental section is again very simple and mainly composed of small kernels (few instructions). No performance figures have been presented in addition to additional instructions.
Reviewer 2 Report
This paper proposes a VLIW architecture and an instruction duplication technique for soft-error tolerant systems.
The idea is interesting, but this work seems incomplete.
- The proposed architecture should be evaluated in terms of hardware cost, clock frequency, power consumption and so on.
- The algorithm of the duplication technique should be formally described.
- The experiments should be conducted more extensively, by using more realistic applications, changing the hardware parameters (e.g., the number of functional units), and so on.
Reviewer 3 Report
I send the review in the attachment file.
